# Imprinted Photonic Crystal-Film-Based Smartphone-Compatible Label-Free Optical Sensor for SARS-CoV-2 Testing

**DOI:** 10.3390/bios12040200

**Published:** 2022-03-28

**Authors:** Daiki Kawasaki, Hirotaka Yamada, Kenji Sueyoshi, Hideaki Hisamoto, Tatsuro Endo

**Affiliations:** 1Department of Applied Chemistry, Graduate School of Engineering, Osaka Prefecture University, Sakai 599-8531, Japan; syb02029@edu.osakafu-u.ac.jp (D.K.); sxb02146@edu.osakafu-u.ac.jp (H.Y.); sueyoshi@chem.osakafu-u.ac.jp (K.S.); hisamoto@chem.osakafu-u.ac.jp (H.H.); 2Japan Science and Technology Agency (JST), Precursory Research for Embryonic Science and Technology (PRESTO), 5-3 Yonban-cho, Chiyoda, Tokyo 102-8666, Japan

**Keywords:** SARS-CoV-2, coronavirus, optical sensor, biosensor, nanoimprint lithography, smartphone

## Abstract

The coronavirus disease (COVID-19) caused by SARS-CoV-2 has caused a global pandemic. To manage and control the spread of the infection, it is crucial to develop and implement technologies for the early identification of infected individuals and rapid informatization in communities. For the realization of such a technology, a widely available and highly usable sensor for sensitive and specific assay of the virus plays a fundamental role. In this study, we developed an optical sensor based on an imprinted photonic crystal film (IPCF) for quick, simple, and cost-effective detection of SARS-CoV-2 spike protein in artificial saliva. Our IPCF sensor enabled label-free and highly sensitive detection with a smartphone-equipped optical setup. The IPCF surface was functionalized with an anti-SARS-CoV-2 spike protein antibody for immunoassay. We evaluated the specificity and sensitivity of the IPCF sensor for quantitative detection of the spike protein in artificial saliva using simple reflectometry with a spectrometer-equipped optical setup. Specific and quantitative detection of the spike protein was successfully achieved, with a low detection limit of 429 fg/mL. In the demonstration of reflectometric detection with a smartphone-equipped setup, the sensitivity was comparable with that with a spectrometer-equipped setup. The test result is returned immediately and can be saved to cloud storage. In addition, it costs less than USD 1 for one IPCF to be used for diagnosis. Thus, the developed IPCF has the potential to realize a widely available and highly usable sensor.

## 1. Introduction

Since the novel coronavirus (severe acute respiratory syndrome coronavirus 2, SARS-CoV-2) was first confirmed in December 2019, it has spread rapidly across the world, bringing about the coronavirus disease 2019 (COVID-19) pandemic [1,2]. Currently, the pandemic is triggering many researchers to develop biomedical technologies and diagnosis systems to control the spread of the disease [3,4,5]. Biosensors for immediate detection of viruses and diagnosis methods for identification of infected human individuals are the key technologies for the development of medical systems to prevent pandemics [6,7]. In this regard, high-sensitivity and -accuracy biosensors are required for accurate medical diagnosis. Meanwhile, high usability and wide availability of sensors and methods are required as preventative measures against the spread of viruses [8,9,10]. In addition, such sensors should be highly compatible with digital communication systems, accommodating digital transformations in medical systems worldwide [5,10,11].

Several diagnostic sensors and methods have been developed for the detection and quantification of novel coronavirus and its related proteins and genes [11,12]. Genetic tests, which mainly use polymerase chain reaction (PCR), generally show high sensitivity and accuracy, but their high cost, complex equipment and operation, and long time to diagnosis make them unsuitable for early diagnosis [12,13,14]. Alternatively, antigen testing is widely used as an early diagnosis method because of its simplicity and speed of testing [15,16,17]. Antigen testing is based on two types of sensors: labelling and non-labeling. The labelling method requires a complicated procedure for testing [18,19,20], while label-free methods allow for simple operation [21,22,23]. Label-free sensors are good candidates for on-site and quick testing. Recently, many label-free biosensors have demonstrated highly sensitive and rapid detection of target molecules using simple and quick operating devices. Electrochemical sensors, such as field effect transistors (FETs) and piezoelectric devices, have been successfully demonstrated to have high sensitivity and low limit of detection (LOD); however, these devices are complicated and expensive because the electronically functionalized layers of the device generally require vacuum conditions for fabrication [24,25,26]. Optical sensors based on nano-photonics, such as plasmonic nanostructures and photonic crystals, also have high sensitivity. Moreover, these devices are relatively simple and cost-effective owing to their dielectric nanostructures fabricated by nanoimprint lithography [27,28,29,30]. We previously developed cost-effective material-based photonic crystal chips for application to label-free biosensors and successfully demonstrated highly sensitive detection and analysis of biomolecules and viruses with very simple setups [31,32,33]. However, our photonic sensors require a spectrometer and computer with built-in software, which makes it difficult for our sensors to be highly usable and widely available for practical application in social medical systems.

In this study, we developed a polymer-based imprinted photonic crystal film (IPCF) for simple and rapid optical detection and quantification of the SARS-CoV-2 spike protein. Our photonic crystal film is mass-producible using a nanoimprint technique, which realizes low cost (approximately USD 1) to use one chip for diagnosis. Thus, the IPCF could realize a cost-effective diagnosis of COVID-19. The IPCF efficiently diffracts light in an angle-dependent manner and decreases the diffraction intensity due to disturbance from periodic refractive index distributions by adsorbing target molecules on the surface. By immobilizing antibody molecules on the surface, the IPCF enables specific detection of antigens. Therefore, the IPCF makes it possible to sensitively detect and specifically quantify target molecules adsorbed on the IPCF surface at the diffraction limit scale by using reflectometry with a simple and small optical setup. Owing to this optical sensing property, the IPCF can work well with smartphone-based spectroscopic equipment. Smartphone-based measurement is suitable for widely available and highly usable diagnostic tests and can be made compatible with digital medical systems.

In this study, we fabricated and characterized the IPCF sensor and applied it to smartphone-based measurements (Figure 1). First, an anti-SARS-CoV-2 spike antibody was immobilized on the IPCF surface. Then, the responsivity and specificity of spike proteins in artificial saliva were investigated using a commercially available spectrometer. Because good responsivity and specificity with a high signal-to-noise ratio (SNR) were confirmed by using a buffer solution, spike protein detection in the artificial saliva was investigated. The sensitivity of the IPCF sensor was theoretically evaluated using the finite-difference time-domain (FDTD) method. Finally, the IPCF-sensor-equipped smartphone-based spectroscopic setup was constructed, and its sensing performance was compared to that of a setup with a commercially available spectrometer.

## 2. Materials and Methods

### 2.1. Functionalization of the IPCF Surface

A cycloolefin polymer (COP)-based IPCF was used for the sensor chips (Figure 2a). The IPCF had a nanostructured pattern with periodically arrayed nanoholes, a diameter (*d*) of 230 nm, and a lattice constant (*a*) of 460 nm, as shown in the image obtained by scanning electron microscopy (FE-SEM, SU8010, Hitachi, Ibaraki, Japan) (Figure 2b). First, the IPCF chip was air-plasma treated to hydrophilize the surface. Next, the surface was silanized by incubation in a 1 wt% 3-aminopropyltriethoxysilane (APTES) ethanol:water (95:5 wt%) solution for 30 min at 37 °C, followed by washing with ethanol and ultrapure water. Then, the salinized IPCF was incubated in a 1 wt% glutaraldehyde (GA) aqueous solution for 1 h, followed by washing with ultrapure water. Then, the anti-SARS-CoV-2 spike antibody was immobilized on the IPCF surface by incubating the aldehyde-functionalized IPCF in 1 ng/mL antibody-containing phosphate buffered saline (PBS) solution (pH 7.4) for 1 h [34], followed by washing with PBS solution and ultrapure water. To prevent nonspecific antigen binding to aldehyde groups left on the surface, the antibody-immobilized IPCF was incubated in 100 mM ethanolamine (EA)-containing PBS solution for 30 min to block aldehyde groups and introduce hydroxyl groups. All incubation processes were performed at 37 °C. The COP film (FLH230/200-120) was purchased from Scivax Co. Ltd., Kanagawa, Japan. APTES was purchased from Tokyo Chemical Industry Co. Ltd., Tokyo, Japan. The 25% GA solution was purchased from Wako Pure Chemical Co., Osaka, Japan. The PBS solution was purchased from Nacalai Tesque Inc., Kyoto, Japan.

### 2.2. Optical Setup and Measurement

The optical measurements were based on reflectometry with a simple and compact setup, which consisted of a tungsten-halogen lamp light source (LS-1, Ocean Insight Tokyo, Japan), optical fiber (R400-7, UV-VIS, Ocean Insight) and a spectrometric part. In this study, two spectrometric setups were used: one was composed of a commercially available spectrometer (FLAME UV-VIS, Ocean Insight) and operation software (Ocean view, Ocean Insight) and the other of a slit, grating (Go spectro, KLV Co. Ltd., Tokyo, Japan), CMOS sensor equipped with a smartphone (iPhone 11, Apple Inc., Los Altos, CA, USA) and application software (Go spectro). In the smartphone-based spectroscopy, the color spectrum of LED light obtained by the CCD sensor was used to calibrate the wavelength in the measured color spectrum. The reflection spectra of the IPCF were measured and normalized to that of the IPCF after blocking treatment with EA, described as functionalized-IPCF (F-IPCF). The maximum reflection intensity was defined as *R*, and the label-free detection of the target molecules was based on the decrease in the reflection intensity, ΔR. The reflectometry was conducted in air after each incubation process.

### 2.3. Label-Free Detection of SARS-CoV-2 Spike Proteins

Artificial saliva (A. saliva), a PBS solution containing 200 μg/mL human serum albumin (HSA), 20 μg/mL human-derived immunoglobulin G (IgG), and 20 mg/mL mucin were prepared [35]. Then, the human-cell-derived native SARS-CoV-2 spike-protein-containing PBS or A. saliva solution (1 pg/mL–100 ng/mL) was prepared. To investigate the specificity of the target proteins, PBS solutions containing HSA (200 μg/mL), IgG (20 μg/mL), or mucin (20 mg/mL) were also prepared. The decrease in reflection intensity, ΔR, is the response between the reflection intensities of the F-IPCF before and after incubation in the sample solution for an arbitrary time (*t* min) at 37 °C. In the investigation of specificity and sensitivity, the incubation time, *t*, was 60 min. In the experiment on responsivity related to incubation time, the incubation time, *t*, was investigated at 30, 60, and 120 min. Triplicate measurement using different three sensor chips was performed for each experiment. IgG, HSA, and mucin were purchased from Sigma-Aldrich Japan Inc., Tokyo, Japan. Spike proteins and anti-spike protein antibody were purchased from Funakoshi Co. Ltd., Tokyo, Japan.

## 3. Results and Discussions

### 3.1. IPCF for Label-Free Optical Sensor

To apply the IPCF to a label-free optical sensor of the SARS-CoV-2 spike protein (S) using an antigen–antibody binding reaction, anti-SARS-CoV-2 spike protein antibody was first modified on the IPCF surface. In this study, a COP-based IPCF was used for the sensor chip (Figure 2a). As shown in the SEM image (Figure 2b), the IPCF had a nanostructured pattern with periodically arrayed nanoholes. The diameter (*d*) and lattice constant (*a*) were 230 nm and 460 nm, respectively. Label-free optical sensing using the IPCF was based on reflectometry with a simple and compact optical setup composed of a white light source, optical fiber probe, and spectrometric system. A commercially available spectrometer and operation software were used for the characterization of the IPCF sensor (Figure 2c).

The normalized reflection spectrum of the IPCF is shown in Figure 2d. The IPCF surface diffracts incident light according to Bragg diffraction (2asinθ=mλ). The 2nd (*m* = 2) peak was intensified by the reflected light on the bottom surface of the film, and the 2nd peak intensity was higher than that of the 1st and 3rd peaks (Figure 2d). The IPCF surface was chemically functionalized for application to optical sensors (see Section 2.1, Appendix A). The air-plasma-treated IPCF was then modified with APTES and GA, followed by anti-*S* antibody (1 μg/mL) modification on the surface of the IPCF. To prevent nonspecific binding of any proteins to the aldehyde groups left on the surface, these groups were blocked by ethanolamine. Finally, the F-IPCF was obtained (Appendix A). Then, the label-free detection of the spike protein (*S*) was conducted with the F-IPCF based on an antigen–antibody reaction (Figure 3a). The reflection spectrum of the IPCF after each modification process was measured, and the maximum reflection intensity (*R*) was plotted (*N* = 3) (Figure 3b). The reflection spectrum of each IPCF was normalized to that of the F-IPCF. As shown in the spectra in Figure 2f, the reflection intensity *R* decreased as the IPCF surface was modified. This decrease in *R* was attributed to the increase in the nanoscale surface roughness by the chemical coatings on the IPCF surface. Appendix A shows SEM images of the bare IPCF, air-plasma treated IPCF, and GA-modified IPCF. The diameter (*d*) of the holes was increased by air-plasma treatment, and then *d* was reduced by the APTES-GA coating. In addition, the response of the F-IPCF to the spike proteins (*S*) (1 ng/mL in PBS) was confirmed by the decrease in reflection intensity. The response to spike proteins (*S*) was considered to arise from the scattering of the diffracted light by the spike proteins (*S*) on the nanohole surface. In addition, the low cost and quick producibility make it possible for the IPCF sensor to be practically used for cost-effective antigen testing in medical diagnosis.

### 3.2. Specific Detection of SARS-CoV-2 Spike Proteins

Considering the application of the IPCF to antigen testing as an early and accurate diagnostic method, specificity is one of the most important properties. In the practical use of the IPCF sensor for on-site diagnosis, samples would be human saliva, sweat, or urine. In this part, the specificity of the F-IPCF to SARS-CoV-2 spike proteins (*S*) in A. saliva is evaluated. The A. saliva contained 200 μg/mL HSA, 20 μg/mL IgG, and 20 mg/mL mucin in PBS (pH 7.4) solution. Label-free detection of spike proteins was based on the decrease in reflection intensity, ΔR, of the F-IPCF from before to after incubation in the sample solution for 1 h (Figure 4a,b). Figure 4c shows the responses to the A.-saliva-containing contaminants, A. saliva, and 1 ng/mL spike proteins (*S*) in PBS and A. saliva. The F-IPCF did not respond to inorganic chemicals in PBS, whereas it slightly responded to contaminant proteins due to the nonspecific adsorption of the contaminants to the F-IPCF surface. This responsivity to contaminants was relatively low to specifically detect the spike proteins, where the ratio of responsivity to A. saliva to that of *S* in A. saliva was nearly 0.1 ($\Delta R_{A.\;saliva}/\Delta R_{S\;in\;A.\;saliva}≃\;0.1$). It was expected that the lower responsivity to *S* in A. saliva than that to *S* in PBS was caused by several factors regarding the difference in the spike-protein-containing solutions: nonspecific adsorption of contaminants or spike proteins, the binding constant of spike proteins to antibody, and the kinetics of the binding reaction.

### 3.3. Characterization of the IPCF Sensor

To apply the IPCF sensor for accurate diagnosis, its quantitativity and sensitivity were investigated. The response of the IPCF sensor to spike proteins with a concentration range from 1 pg/mL to 100 ng/mL in PBS or A. saliva solution was investigated, and then the calibration property was evaluated. Figure 5a,b shows the reflection spectra of the IPCF sensor after incubation for each spike-protein-containing PBS (Figure 5a) and A. saliva (Figure 5b) solution. Figure 5c,d show plots of the response to each spike concentration. Figure 5e,f displays the calibration curves to the spike protein concentration, and the LOD based on three times the standard deviation (3SD) was 1.2 pg/mL in the case of PBS and 429 fg/mL in the case of A. saliva, which were calculated from the results of triplicate measurements with independent IPCF chips (*N* = 3). The LOD for A. saliva was lower than that for the PBS solution. The relative standard deviation (RSD) in the case of PBS was higher than that of A. saliva (Appendix A), while good linearities were confirmed in both PBS and A. saliva samples (R^2^ = 0.9931 in the PBS sample and R^2^ = 0.9870 in A. saliva).

Figure 6a shows plots of the responses to molecular concentrations, *C* (nM), of spike proteins, derived from the plots in Figure 5a,b. Figure 6b shows the slopes of the calibration curves in the cases of PBS and A. saliva, which indicates the sensitivity. The sensitivity in the case of PBS was higher than that of A. saliva, which is in agreement with the results in Figure 4c. To investigate the difference in sensitivity, the binding constant *Ka* (M^−1^) was evaluated by the adsorption isotherm, which was applied to the IPCF sensor. The Langmuir adsorption isotherm in a solution is described as
(1)$$\frac{C}{W}=\frac{C}{W_{max}}+\frac{1}{W_{max}\cdot Ka}$$
where *C* is the concentration of adsorbates, W is the adsorption amount of the adsorbates, and $W_{max}$ is the saturated adsorption amount. To apply the Langmuir adsorption isotherm to the IPCF surface adsorption, W and $W_{max}$ were replaced by ΔR and $\Delta R_{max}$, where the ΔR value for a 100 ng/mL sample concentration was used as $\Delta R_{max}$. Thus, the applied adsorption isotherm is described as
(2)$$\frac{C}{\Delta R}=\frac{C}{\Delta R_{max}}+\frac{1}{\Delta R_{max}\cdot Ka}.$$

Figure 6c shows the applied adsorption isotherm, and Figure 6d represents the logarithmic binding constants derived from Equation (2) using PBS and A. saliva as the sample solutions. The estimated binding constant for PBS was higher than that for A. saliva. To study the response in the different sample solutions, the responses to various incubation times (*t* = 30, 60, and 120 min) were evaluated. In this experiment, the concentration of spike proteins was 1 ng/mL. Figure 6e displays plots of the responses to each incubation time and fitted curves of the applied first-order equation, which is described as
(3)$$\Delta R=\Delta R_{t=120}\left\{1-\exp\left(-ka\cdot t\right)\right\}$$
where ka is the binding rate constant [36]. It was considered that the response to spike proteins was saturated at an incubation time of 120 min. Figure 6f shows the response at *t* = 120 min for PBS and A. saliva. The saturated response in the case of PBS was higher than that in the case of A. saliva, which suggests that more spike proteins nonspecifically adsorbed to the IPCF surface in the case of PBS than in the case of A. saliva. The estimated binding constants, binding rate, and dissociation rate constants are listed in Table 1. Both the binding constant and binding rate in the case of PBS were higher than those in A. saliva. However, when the sample concentration was 1 pg/mL, which was the lowest concentration tested in the calibration experiment, almost the same response was confirmed between PBS and A. saliva sample solutions. Considering the saturated response described above, it was expected that the nonspecifically adsorbed contaminants in A. saliva would increase the response in the low spike concentration range. However, the adsorption sites for contaminants, which were contained in sample solutions with a constant concentration independent of the target concentration, were also constant. It was expected that the nonspecifically adsorbed contaminants were constant in their amounts regardless of the spike protein concentration and increased apparent response as a background for the detection of targets. Meanwhile, in the case of PBS without contaminants, the amount of nonspecifically adsorbed spike proteins would increase with increasing concentration in the sample solutions, resulting in an increase in the apparent response to total spikes, both specifically and nonspecifically adsorbed to the IPCF surface. It was considered that this increase in nonspecific adsorption of spike proteins would increase the RSD as the spike concentration increased. The nonspecifically adsorbed molecules would decrease their apparent binding constant estimated by the applied Langmuir adsorption isotherm; thus, the net binding constant of the spike proteins in PBS solution was higher than the apparent value. To summarize the sample solution in which the IPCF sensor works, it was indicated that the nonspecifically adsorbed contaminants in A. saliva decreased the binding constant and rate of the antigen–antibody reaction of spike proteins, thus decreasing the sensitivity and RSD. Thus far, the IPCF has been characterized from the perspective of a sensor for SARS-CoV-2, and it was expected that the IPCF sensor would have the potential for sensitive and specific detection and quantification of the virus using a compact optical setup. However, because the spike protein was used to demonstrate the IPCF sensor, the differences between spike protein and virus should be discussed (see Appendix A). The spike protein generally conforms to a trimer with a size of approximately 20 nm, while SARS-CoV-2 has 20–30 spikes on its surface, and the particle size is approximately 100 nm [37]. The response of the IPCF sensor to the adsorbates is attributed to the decrease in the diffraction intensity by scattering due to the adsorbates. The scattering intensity by the adsorbates depends on their dielectric characteristics, size, and surface density. According to the theoretical evaluation of the responsivity of the IPCF sensor to dielectric particles on the surface, it is expected that the IPCF sensor is more sensitive to the virus particle than the spike proteins. Thus, it is expected that the IPCF sensor can be applied to the highly sensitive testing of the virus in human saliva.

### 3.4. Smartphone-Based Optical System

Considering the wide availability and high usability of digital medical communication systems, such as point-of-care (POC) systems, it is favorable for the IPCF sensor to be equipped with widely available instruments, such as smartphones. In this study, we developed a smartphone-based (iPhone 11, Apple, USA) spectrometric system to demonstrate its reflectometric detection of spike proteins. The smartphone-based spectrometer was composed of a slit, grating, and CMOS sensor of the smartphone. The spectrum was displayed on the screen of the smartphone using application software (Go spectro) (Figure 7a,b). The wavelength calibration of the spectrum image obtained by the CMOS was based on the spectrum of the fluorescent lamp, which ensured the accuracy of the wavelength corresponding to the measured spectrum image. The reflection spectrum of the bare IPCF using the smartphone-based setup is shown in Figure 7c. The reflection spectrum obtained by the spectrometer-based setup is also shown in Figure 7c. The shift of the peak wavelength (15.6 nm) was attributed to the low calibration accuracy and wavelength resolution of the smartphone-based spectroscopic system. The response to the spike proteins in A. saliva by the smartphone-based system is shown in Figure 7d. The response is described as a decrease in the intensity of the reflection peak, as shown in Figure 7d. Then, the smartphone-based spectrometric system was compared with the spectrometer in terms of response to the spike proteins (1 ng/mL) contained in A. saliva and the RSD (Figure 7e,f). The smartphone-based spectrometric system was not inferior to the spectrometer-based system with respect to response, ΔR, to the target molecules. However, the RSD of the smartphone-based system was 2.8 times higher, which would lead to a lower diagnostic accuracy than that of the spectrometer-based system. The response is attributed to the light sensitivity of the CMOS; meanwhile, its dispersion is affected by the wavelength resolution and calibration accuracy, which are attributed to the optics and pixels of the CMOS. Thus, it is expected that the RSD can be reduced by improving the optical components, such as using a narrower slit, grating with more grooves, and more robust mounts. Thus, the required specifications can be realized for a smartphone-based system with noninferiority to the spectrometer-based system. The operation software (Go spectro) enables real-time monitoring and quick display of the results, saving the data to the cloud. This data analysis and informative system has more advantages in the medical diagnosis of pandemics like COVID-19 than present diagnostic tests, such as PCR or paper-based antigen tests.

## 4. Conclusions

In this study, we developed a label-free optical sensor for SARS-CoV-2 based on an IPCF and demonstrated sensitive and specific detection of spike proteins using a simple and quick reflectometric method. The IPCF has potential for application in cost-effective and simple sensing systems thanks to its Bragg’s diffraction. It costs less than USD 1 to fabricate one chip and takes assay reaction time less than 60 min. The low limit of detection (LOD) for spike proteins in A. saliva (429 fg/mL) was achieved. To summarize, the IPCF sensor has potential for practical use in medical diagnosis to prevent the spread of infectious disease as a quick, cost-effective, and highly sensitive antigen testing platform, as shown in Table 1. For practical application of the IPCF sensor in digitally transformed medical diagnosis systems in the near future, we investigated a smartphone-based measurement system using the IPCF sensor and compared it with a commercially available spectrometer-based system. The smartphone-based spectrometric system had a sensitivity comparable to that of a commercially available spectrometric system. Thus, the smartphone-based system can be a good label-free optical sensing platform with noninferiority to spectrometer-based systems. Therefore, the IPCF sensor is expected to be suitable for widely available and highly usable antigen testing with smartphones and can be used in digital medical diagnosis systems, such as POC [9]. On the other hand, external light source and optical fiber make the mobile feature of smartphone discount for on-site detection. In the future, we should develop the optical device integrated with the smartphone. Specifically, the smartphone-internal LED can replace the external light source, or some image analyzing methods might make the spectrometric method unnecessary. In this study, we developed the IPCF sensor for SARS-CoV-2, but it could be applied to any antigen by appropriate chemical functionalization of the surface.

## Figures and Tables

**Figure 1 biosensors-12-00200-f001:**
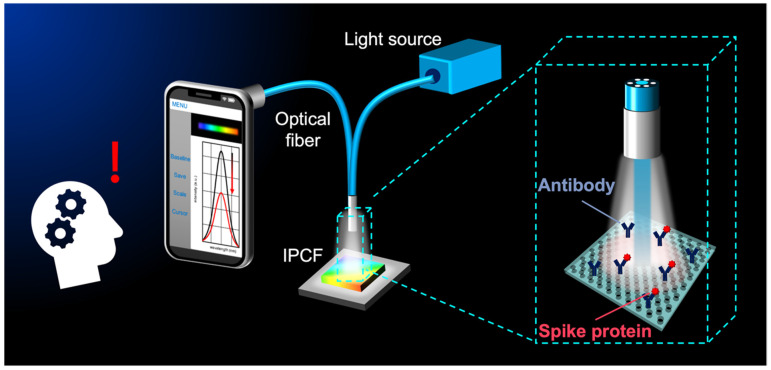
Conceptual illustration for smartphone-based optical detection of SARS-CoV-2 spike proteins using the imprinted photonic crystal film (IPCF) sensor.

**Figure 2 biosensors-12-00200-f002:**
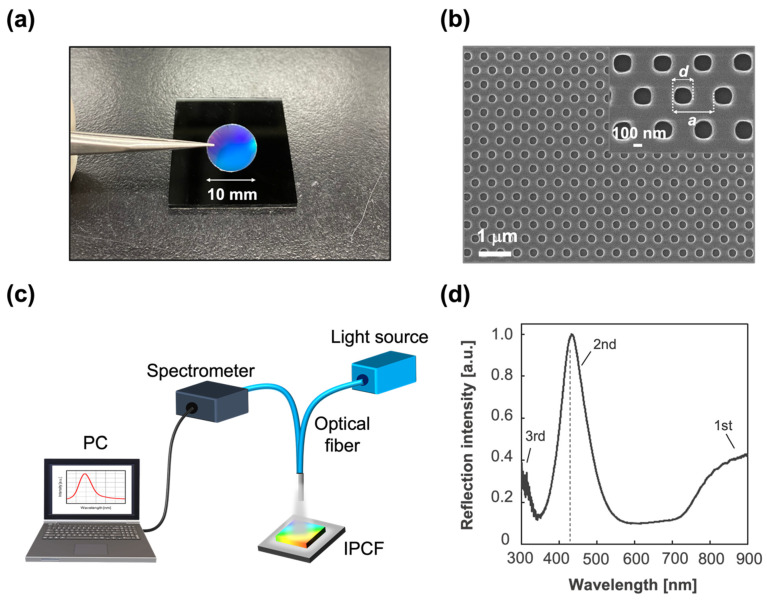
IPCF for label-free optical sensor. (**a**) Image of an IPCF chip. The diameter of the chip used for the sensor was 10 mm. (**b**) SEM image of the IPCF surface. The inset image shows that the diameter of the holes (*d*) was 230 nm and the lattice constant (*a*) was 460 nm. (**c**) Schematic illustration of the optical setup for reflectometry with the IPCF. It was composed of a tungsten-halogen lamp as a light source, an optical fiber probe, and a spectrometer. (**d**) Typical reflection spectrum of the IPCF. The diffraction was attributed to Bragg’s law, 2asinθ=mλ; the observed peaks were distributed to the 1st, 2nd, and 3rd Bragg peaks (*m* = 1, 2, and 3).

**Figure 3 biosensors-12-00200-f003:**
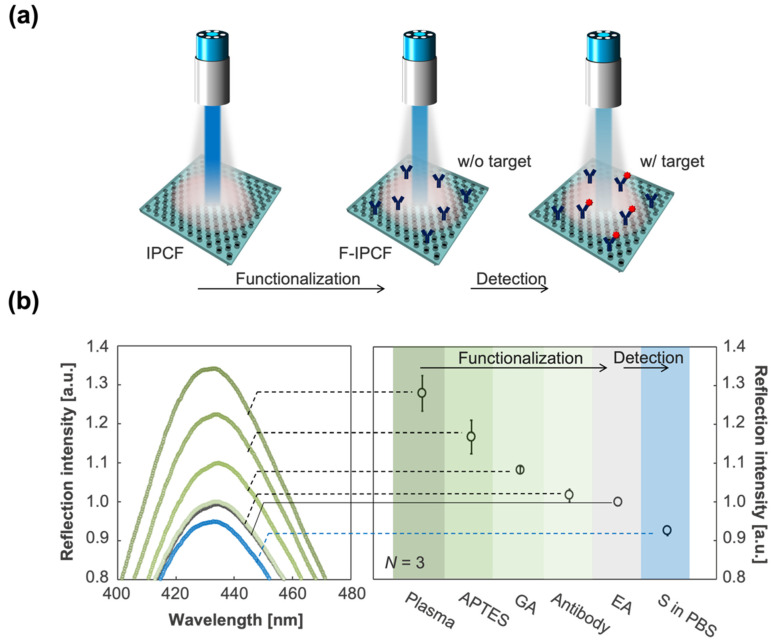
Functionalization of the IPCF for application to label-free optical sensor of SARS-CoV-2 spike proteins (*S*). (**a**) Schematics of the functionalization of the IPCF for label-free detection of spike proteins (*S*). (**b**) (Left) Magnified reflection spectra for the 2nd peak of IPCF during the modification processes: air-plasma treatment, APTES, GA, antibody and EA modifications, and after incubation in the spike-protein-containing (1 ng/mL) PBS sample solution. Each spectrum was normalized by that of F-IPCF. (Right) Plots of maximum reflection intensity, *R*, after each modification process (*N* = 3).

**Figure 4 biosensors-12-00200-f004:**
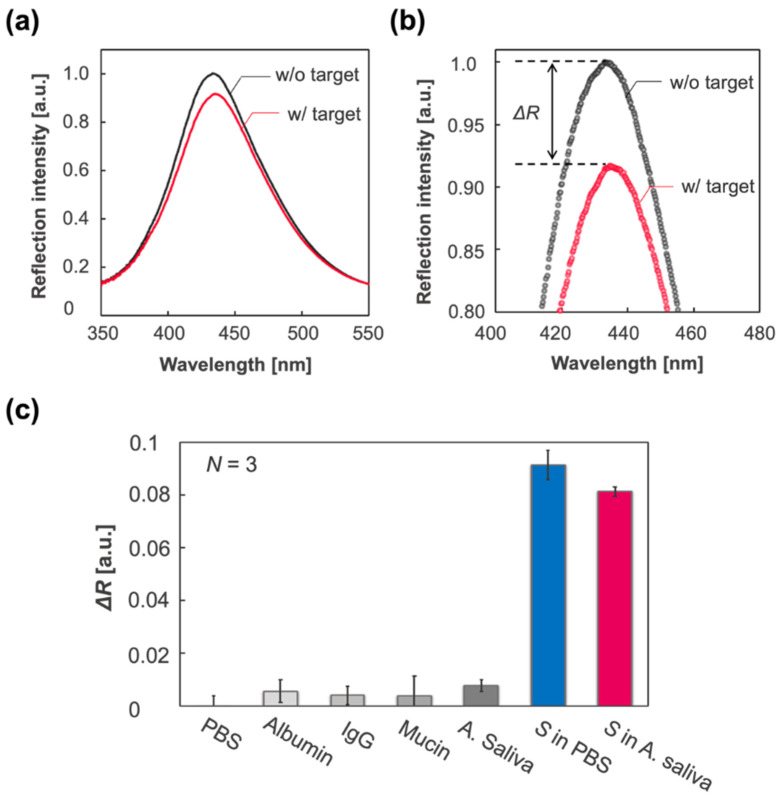
Label-free specific detection of SARS-CoV-2 spike proteins (*S*). (**a**) Reflection spectrum of the IPCF before (F-IPCF, black) and after (Target, magenta) incubation in A. saliva sample solution containing spike proteins (1 ng/mL). (**b**) Enlarged spectra around the peaks. The label-free detection of the target (*S*) was based on the decrease in reflection intensity, ΔR. (**c**) Selectivity for the spike proteins. From left to right: PBS (pH 7.4), albumin (200 μg/mL in PBS), IgG (20 μg/mL in PBS), mucin (20 mg/mL in PBS), A. saliva containing albumin, IgG, and mucin in PBS (grey scale bars), spike-protein-containing PBS (blue bar), and A. saliva (magenta bar).

**Figure 5 biosensors-12-00200-f005:**
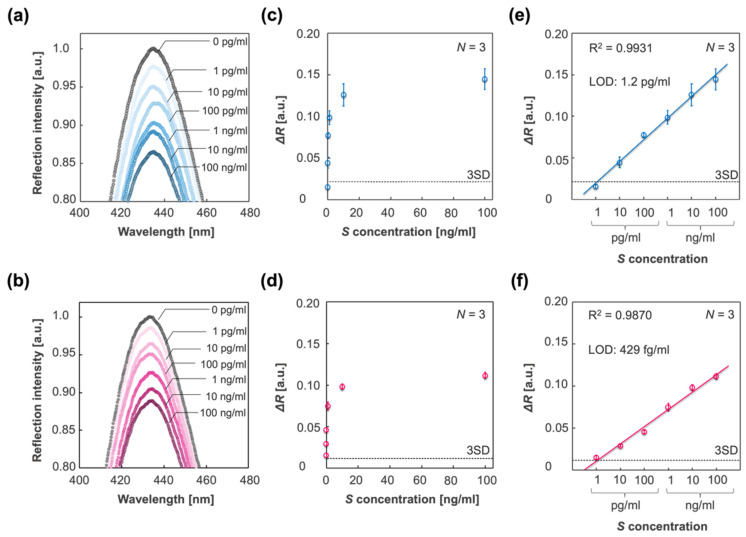
Detection and quantification of spike proteins in different sample solutions. (**a**,**b**) Reflection spectra of the IPCF before (black plots) and after incubation in respective sample solutions containing spike proteins with different concentrations of 1 pg/mL to 100 ng/mL: PBS (blue scaled plots) and A. saliva (magenta scaled plots). (**c**,**d**) Plots of responses to each spike protein concentration in the cases of PBS (**c**) and A. saliva (**d**). (**e**,**f**) Calibration curve of the response to spike protein concentration in the cases of PBS (**e**) and A. saliva (**f**). The three times standard deviation (3SD) lines are displayed, and the limit of detections (LOD) based on the 3SD values are described. Triplicate measurements with independent chips were conducted (*N* = 3).

**Figure 6 biosensors-12-00200-f006:**
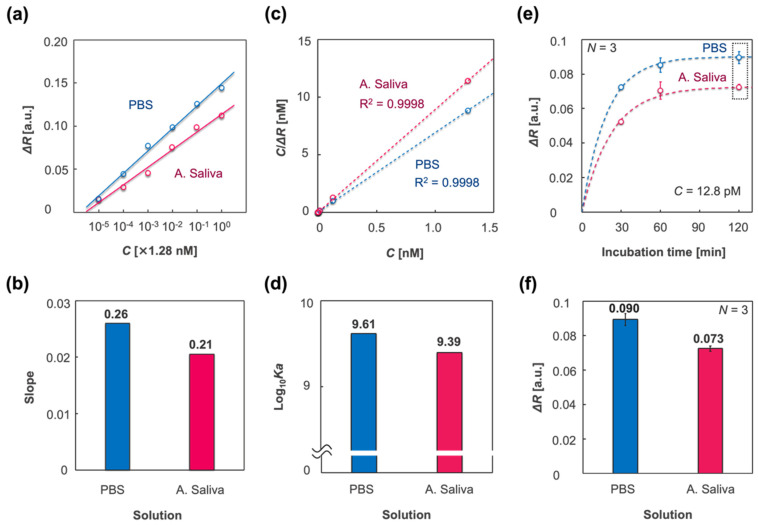
Characterization of the IPCF surface adsorption using different sample solutions: PBS (blue) and A. saliva (magenta). (**a**) Plots of responses to spike protein concentration *C* (M), which was derived from the plots in Figure 5e,f. (**b**) Slopes of the calibration curves in (**a**). (**c**) Langmuir adsorption isotherm applied to the IPCF surface, which was described as $C/\Delta R=C/\Delta R_{max}+1/(\Delta R_{max}Ka)$, where $\Delta R_{max}$ is the response when *C* was 1.28 nM and *Ka* is the binding constant. (**d**) Logarithmic binding constants estimated by applied adsorption isotherm. (**e**) Plots of responses to incubation time, *t* = 30, 60, 120 (min)., where the fitting curves are described by a first-order equation: $\Delta R=\Delta R_{t=120}\left\{1-\exp\left(-ka*t\right)\right\}$, where *ka* is the binding rate constant. (**f**) Saturated response (*t* = 120 min) derived from the plots in the dashed square in (**e**). The values are described above the bars. Triplicate measurements were conducted with independent chips (*N* = 3).

**Figure 7 biosensors-12-00200-f007:**
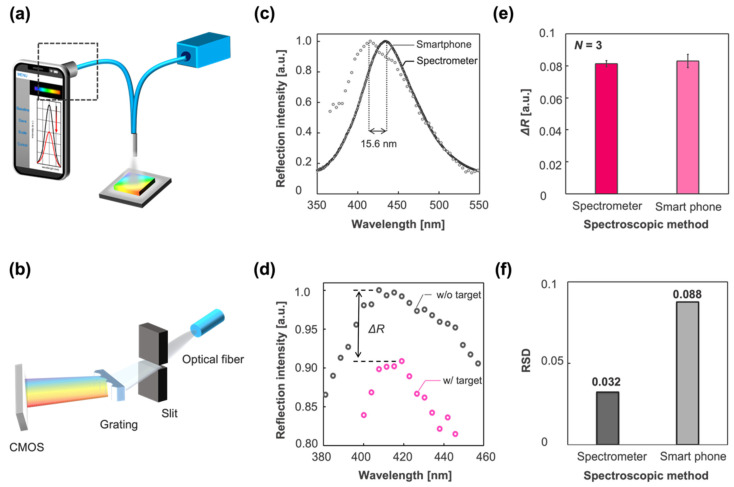
Smartphone-based spectrometric system vs. spectrometer-based system. (**a**) Schematic illustration of the smartphone-based optical setup. (**b**) Smartphone-based spectrometric part indicated by square dashed line in (**a**). It was composed of a slit, grating, and CMOS sensor of the iPhone 11. (**c**) Reflection spectrum of the IPCF obtained by the spectrometer-based system (black circle) and smartphone-based system (gray circle). The peak wavelength was different by 15.6 nm in this case. (**d**) Reflection spectra of the IPCF sensor (gray circle) and those after incubation in spike-protein-containing (1 ng/mL) A. saliva sample solution (pink circle). (**e**) Response to the spike protein in A. saliva (1 ng/mL) measured by the spectrometer-based system (magenta bar) and smartphone-based system (pink bar) (*N* = 3). (**f**) Relative standard deviation (RSD) of the response in (**e**) (black bar: spectrometer; gray bar: smartphone).

**Table 1 biosensors-12-00200-t001:** Comparison of nanomaterial-based immunosensors for Coronavirus detection.

No. [Ref]	Biosensor Platform	Material	Biomarker	Assay Duration	Dynamic Range	LOD
**1. [This work]**	**2D-photonic crystal (Label-free)**	**Imprinted polymer PhC**	**Spike protein S1**	**<15 min**	**0.001–100 ng/mL**	**429 fg/mL**
2 [25]	Field effect transistor (Label-free)	Graphene monolayer	Spike protein S1	Real-time	0.001–10 pg/mL	1 fg/mL
3 [36]	Surface plasmon resonance (Label-free)	Plasmonic nanohole/AuNPs	SARS-CoV-2 Virus	<15 min	0–10^7^ vp/mL	370 vp/mL
4 [38]	Opto-microfluidic chip (Label-free)	Gold nanospikes	Antibodies	30 min	0.1–10,000 ng/mL	0.008 ng/mL
5 [39]	Magnetic-beads-based biosensor (Label-based)	Magnetic beads and carbon black-based electrodes	Spike proteins S1	30 min	0.01–10 μg/mL	19 ng/mmL
6 [40]	Amperometric (Label-based competitive)	AuNPs modified carbon electrodes	Spike protein S1 (MERS)	20 min	0.001–100 ng/mL	0.4 pg/mL

## Data Availability

Data are contained within the article and Appendix A.

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
