# Peer review of "Imprinted Photonic Crystal-Film-Based Smartphone-Compatible Label-Free Optical Sensor for SARS-CoV-2 Testing"

_biosensors, 2022, doi:10.3390/bios12040200_

Round 1
Reviewer 1 Report
It is a very interesting study, and very well performed to rapidly detect SARS-CoV-2 via a polymer-based imprinted photonic crystal film. But the authors need to address the questions listed below before the publication is fully warranted.
1. IPCF abbreviation in abstract is not standard, it is best consistent with the introduction. “Imprinted Photonic Crystal Film (IPCF)”.
2. In Figure 5 (c, d) Plots of responses to each spike protein concentration in the cases of PBS (c) and A. saliva (d). The error in PBS is significantly larger than that in A. saliva. In Figure 5 (e,f), the R2 of calibration curve in PBS is also larger than that in A. saliva. In general, PBS without contaminants and A. saliva with contaminants. What do you think is the main reason for the above difference?
Author Response
Response to the reviewer 1
Thank you very much for your comments and advice. Our comments to your revisions are following.
- IPCF abbreviation in abstract is not standard, it is best consistent with the introduction. “Imprinted Photonic Crystal Film (IPCF)”.
Thank you for your advice. We have revised it. Please see abstract, line 6.
- In Figure 5 (c, d) Plots of responses to each spike protein concentration in the cases of PBS (c) and A. saliva (d). The error in PBS is significantly larger than that in A. saliva. In Figure 5 (e,f), the R2 of calibration curve in PBS is also larger than that in A. saliva. In general, PBS without contaminants and A. saliva with contaminants. What do you think is the main reason for the above difference?
Thank you for your valuable question. The calibration curve was considered to be somewhat like a sigmoidal curve, which was shown in the results in A. saliva. However, in PBS, the response was considered to be excessive in the high concentration range, resulting in apparently higher linearity of calibration curve than that in PBS.

Reviewer 2 Report
Title: Imprinted Photonic Crystal Film-based smartphone-compatible Label-Free Optical Sensor for SARS-CoV-2 Testing.
Minor revision Need
- In Figure 5f, the authors must check the error bar in this calibration plot.
- In figure 6a, the authors must check the error bar in this calibration plot.
- Authors must check the selectivity of the sensor?
- There are many grammatical and typographical errors. Please check the manuscript and refine it carefully.
- Authors should check all figures and captions in the manuscript.
- Figure quality needs to improve.
Author Response
Response to the reviewer 2
Thank you very much for your comments and advice. Our comments to your revisions are following.
- In Figure 5f, the authors must check the error bar in this calibration plot.
Thank you for your advice. We have checked that.
- In figure 6a, the authors must check the error bar in this calibration plot.
Thank you for your advice. This plot has not included error bars for visibility.
- Authors must check the selectivity of the sensor?
Thank you for your question. The selectivity for spike proteins in contaminants of A. saliva is shown in fig. 4c. The selectivity for another viruses like MERS should be investigated in the near future.
- There are many grammatical and typographical errors. Please check the manuscript and refine it carefully.
Thank you for your advice. We have checked and revised it.
- Authors should check all figures and captions in the manuscript.
Thank you for your advice. We have checked and rearranged that.
- Figure quality needs to improve.
Thank you for your advice. We have improved it.

Reviewer 3 Report
Recommendation: Minor revision
The authors demonstrated an imprinted photonic crystal film-based smartphone-compatible sensor for testing of SARS-CoV-2 spike protein in artificial saliva. The authors firstly evaluated the specificity and sensitivity of the sensor for quantitative detection of the spike protein with a spectrometer-equipped optical setup. They successfully detected the spike protein with a low detection limit of 429 fg/ml. Then, they showed that the sensitivity of a smartphone equipped setup was comparable to the sensitivity measured by the spectrometer-equipped setup. Overall, the topic itself is interesting, and the work is quite thorough. The authors may consider the reviewer's comments before the paper can be accepted.
Detailed comments:
1) The smartphone-based spectroscopy for biosensing is sound. However, the proposed setup in this manuscript still depends on the optical fiber and light source, which are not widely available for POC detection. Can the authors comment on how these components can be integrated in the smartphone-based system?
2) Materials and Methods: The fabrication of the imprinted photonic crystal film should be provided.
3) Fig. 2c and Fig. 7a: Why are the probe tips (gray color) mismatched with the optical fibers (blue color)?
4) Fig. 4c: It is not clear how the authors conducted the triplicate measurements for different sample solutions. Did the authors test different sensors? How about the consistency among different sensors?
Author Response
Response to the reviewer 3
Thank you very much for your comments and advice. Our comments to your revisions are following.
1) The smartphone-based spectroscopy for biosensing is sound. However, the proposed setup in this manuscript still depends on the optical fiber and light source, which are not widely available for POC detection. Can the authors comment on how these components can be integrated in the smartphone-based system?
Thank you for your valuable question. We have added description about that in conclusion part. Please see line 380-382.
2) Materials and Methods: The fabrication of the imprinted photonic crystal film should be provided.
Thank you for your advice. In this report, we have used the commercially available photonic crystal film, which was investigated in our previous work [32].
3) Fig. 2c and Fig. 7a: Why are the probe tips (gray color) mismatched with the optical fibers (blue color)?
Thank you for your question. It is actually not mismatched, but it appears to be mismatched by lighting in the figure.
4) Fig. 4c: It is not clear how the authors conducted the triplicate measurements for different sample solutions. Did the authors test different sensors? How about the consistency among different sensors?
Thank you for your valuable question. We used different three sensor chips for triplicate measurements. Thus, the error bar in fig. 4c shows the consistency among sensor chips.
We have added explanation about this. Please see line 142-143.
